# Advancement in *Salmonella* Detection Methods: From Conventional to Electrochemical-Based Sensing Detection

**DOI:** 10.3390/bios11090346

**Published:** 2021-09-18

**Authors:** Mohd Syafiq Awang, Yazmin Bustami, Hairul Hisham Hamzah, Nor Syafirah Zambry, Mohamad Ahmad Najib, Muhammad Fazli Khalid, Ismail Aziah, Asrulnizam Abd Manaf

**Affiliations:** 1Collaborative Microelectronic Design Excellence Centre (CEDEC), Sains@USM, Universiti Sains Malaysia, Level 1, Block C, No. 10 Persiaran Bukit Jambul, Bayan Lepas 11900, Malaysia; mohdsyafiqawang92@student.usm.my; 2School of Biological Sciences, Universiti Sains Malaysia (USM), Gelugor 11800, Malaysia; ybustami@usm.my; 3School of Chemical Sciences, Universiti Sains Malaysia (USM), Gelugor 11800, Malaysia; hishamhamzah@usm.my; 4Institute for Research in Molecular Medicine (INFORMM), Health Campus, Universiti Sains Malaysia (USM), Kubang Kerian 16150, Malaysia; norsyafirah@usm.my (N.S.Z.); najib@student.usm.my (M.A.N.); fazlikhalid@usm.my (M.F.K.)

**Keywords:** *Salmonella*, electrochemical biosensor, aptamer, aptasensor, nanomaterials

## Abstract

Large-scale food-borne outbreaks caused by *Salmonella* are rarely seen nowadays, thanks to the advanced nature of the medical system. However, small, localised outbreaks in certain regions still exist and could possess a huge threat to the public health if eradication measure is not initiated. This review discusses the progress of *Salmonella* detection approaches covering their basic principles, characteristics, applications, and performances. Conventional *Salmonella* detection is usually performed using a culture-based method, which is time-consuming, labour intensive, and unsuitable for on-site testing and high-throughput analysis. To date, there are many detection methods with a unique detection system available for *Salmonella* detection utilising immunological-based techniques, molecular-based techniques, mass spectrometry, spectroscopy, optical phenotyping, and biosensor methods. The electrochemical biosensor has growing interest in *Salmonella* detection mainly due to its excellent sensitivity, rapidity, and portability. The use of a highly specific bioreceptor, such as aptamers, and the application of nanomaterials are contributing factors to these excellent characteristics. Furthermore, insight on the types of biorecognition elements, the principles of electrochemical transduction elements, and the miniaturisation potential of electrochemical biosensors are discussed.

## 1. Introduction

*Salmonella* infections cause various morbidity and mortality globally, especially in developing countries [1]. The prevalence is high in low-resource regions, where poor sanitation and lack of clean water supply are an issue [2]. Enteric fever (typhoid and paratyphoid fever) is the febrile illness caused by *Salmonella enterica* serotype Typhi and Paratyphi infection, characterised by a long persistent fever and other gastrointestinal complications. Late treatment or misdiagnosis could increase the risk of death and can lead to other problems such as antibiotic resistance [3].

To date, there are various types of *Salmonella* detection techniques available with unique mechanisms of detection. They are immunological-based assays, such as enzyme-linked immunosorbent assay (ELISA), latex agglutination method, and immunochromatography assay [4,5,6]; molecular-based assays, e.g., polymerase chain reaction (PCR), loop-mediated isothermal amplification (LAMP), nucleic acid sequence-based amplification (NASBA), recombinase polymerase amplification (RPA), DNA microarrays, whole genome-sequencing (WGS) [7,8,9,10,11,12]; mass spectrometry-based methods, e.g., matrix-assisted laser desorption ionisation-time-of-flight mass spectrometry (MALDI-TOF MS), liquid chromatography-mass spectrometry (LC-MS) [13,14]; spectroscopy-based methods, e.g., Raman spectroscopy, near-infrared (NIR) spectroscopy, and hyperspectral imaging (HSI) [15,16,17], optical phenotyping with light diffraction technology [18], and electrochemical biosensors [19]. These methods can identify and discriminate *Salmonella* down to their serotype level [14,20,21,22,23]. Apart from these detection methods, electrochemical biosensors could potentially be an ideal tool for *Salmonella* detection, since they are capable of detecting the presence of *Salmonella* or their cellular components in just a few hours.

Immunological-based assays and molecular-based assays provide excellent detection of *Salmonella* with high affinity. However, the low stability issue of antibody and high operation costs of PCR were deemed to be the drawback of these methods [23]. These two methods are also prone to cross-reactivity issues among *Salmonella* serovar [24,25,26]. DNA microarray and whole-genome sequencing methods were able to provide detailed information on *Salmonella* genotypes [14,27]. Mass spectrometry methods, spectroscopy methods, optical phenotyping are culture-independent methods as these methods utilised the phenotypic characteristics of *Salmonella*, such as peptides fingerprints, absorbance, and imaging characteristics [23,28].

Electrochemical biosensors were developed to detect a variety of analytes of interest such as glucose, cancer biomarkers, viruses, pathogenic bacteria, and heavy metals, among others. In the case of pathogenic bacteria, current studies have reported extensively on the use of electrochemical biosensors in detecting foodborne pathogens, especially *Salmonella*. Electrochemical biosensors are known for their rapid, highly sensitive, specific detection response, and can be integrated and miniaturised into a biosensor device, such as in a microfluidic platform, which can provide an advantage in point of care (POC) testing [22,29]. Additionally, the introduction of aptamer technology as a recognition element in electrochemical biosensors has undoubtedly enhanced the specificity, selectivity, and stability of biosensors [30]. Moreover, the integration of nanomaterials, such as graphene derivates, carbon nanotubes, metallic nanoparticles, metal oxide nanoparticles, and silica nanoparticles, as the surface modifier, has tremendously improved the limit of detection (LOD), as compared to the electrochemical biosensors with only antibody and molecular probes [31]. Thus, the selection of an exceptional biorecognition element combined with a favourable surface modifier in electrochemical biosensor development could contribute to rapid and highly sensitive POC diagnostic devices, particularly for *Salmonella* detection.

## 2. *Salmonella* and Their Related Diseases

The *Salmonella* bacteria was discovered in 1855 by Theobald Smith from pig intestines that been infected with classical swine fever [32]. Upon discovery, *Salmonella* underwent controversial naming as it was named after Dr. Daniel Elmer Salmon, one of Smith’s co-workers. Later, the Centre for Disease Control and Prevention (CDC) suggested a resolution to the nomenclature issue of *Salmonella* by following the recommendation by the World Health Organisation (WHO) collaborating centre [33]. This system suggested the classification of *Salmonella* into two species namely *Salmonella enterica* and *Salmonella bongori*, which were classified based on their 16S rRNA sequence analysis. *Salmonella enterica* can be further grouped into six subspecies, where these subspecies are denoted with a roman numeral symbol from I to VI [34]. *Salmonella enterica* I can be further classified into typhoidal *Salmonella* (*S*. Typhi and *S*. Paratyphi), which only infect humans, and non-typhoidal *Salmonella* (*S*. Typhimurium and *S*. Enteritidis), which infect both humans and animals [35].

*Salmonella* is a flagellated Gram-negative bacteria, facultative anaerobe characterised by O, H, and Vi antigens [36]. It is famously known as a foodborne pathogen, as most infections are acquired from food sources. They cause salmonellosis, clinically ranging from common gastroenteritis (e.g., diarrhoea, abdominal cramps, and fever) to enteric fevers (typhoid and paratyphoid), which could be a life-threatening febrile illness if left untreated [2,36,37,38]. However, the degree of severity depends on the serotype involved and human host status. Infants or children under five years old, elderly, and immunocompromised persons are among the high-risk groups who are susceptible to *Salmonella* infection [32].

Salmonellosis can be acquired from ingestion of food contaminated with *Salmonella*. The sources of contamination could be from the consumption of undercooked food from infected animals in poultry products or other meats, ready-to-eat foods such as fruits and vegetables that have been contaminated with faeces of infected animals, or water that has been contaminated with faeces of infected people or animals [38]. In urban areas, contaminated water sources contribute significantly to *Salmonella* outbreaks [3]. Contact infection can also be acquired through direct exposure to an object or environment contaminated with faeces containing *Salmonella* [38].

Enteric fever is a potentially fatal systemic illness caused by *Salmonella enterica* subspecies serovars Typhi and Paratyphi A, B and C [3,39]. This illness, which is also known as typhoid or paratyphoid fever, was classified based on the serotype of *Salmonella* bacterium infecting the host. Fatality in cases involving typhoid fever was reported to be 1.89 times higher than in paratyphoid fever. These infections resulted in bacteraemic febrile illness, with prolonged high fever, malaise, and headache [39]. If left untreated, it would lead to gastrointestinal bleeding, intestinal perforation, ileus, septic shock, altered mental states (termed the typhoid state) and, consequently, death [3,39,40,41].

In 2017, it was estimated that 14.3 million typhoid and paratyphoid cases occurred globally with 136,000 estimated deaths. Children and the elderly were among the groups of highest recorded fatality, especially in lower-income countries. *Salmonella enterica* serotype Typhi is a major contributor to enteric fever with 76.3% [39]. Typhoid disease is rare in developed countries, but still prevalent in certain areas of developing countries with limited access to clean water and poor sanitation [42,43]. In Malaysia, it was estimated that 0.5 to 0.7 confirmed cases of *Salmonella* infection per 100,000 population were reported in 2015 [44]. Even though most of the outbreaks are endemic, the cases of both typhoid and paratyphoid fever declined globally by 45% from the year 1990 to 2017. Since there was limited use of typhoid vaccine in typhoid-endemic countries, other factors such as the improvement of water supply and sanitation infrastructure, improved food handling practices, and easy access to antibiotic treatment were likely to be the crucial factors that contributed to this decline [39].

The incidence of antimicrobial resistance among bacterial pathogens is currently at an alarming state which includes concern towards resistance in *Salmonella*. This issue was relatively not new to *Salmonella* species as the first chloramphenicol-resistant *S.* Typhi was discovered two years after it was used to treat patients with enteric fever in 1948 [3]. In the 1980s, multidrug-resistant *S*. Typhi against chloramphenicol, ampicillin, trimethoprim, and sulfonamides has become prevalent, followed by nalidixic-acid-resistant *S.* Typhi and *S.* Paratyphi A in the 1990s [45]. In Asia, the majority of *S.* Typhi and Paratyphi A strains are nalidixic-acid resistant and completely resistant towards ciprofloxacin and extended-spectrum cephalosporins have emerged in some parts of the world [46]. It was reported that the antimicrobial resistance of *Salmonella* was linked with the horizontal gene transfer from the virulent multi-drug resistant *S*. Typhimurium DT 104 through its mobile plasmids to other bacteria [47,48,49]. Thus, early detection of *Salmonella* could help to prevent the spread of resistance genes and the optimal use of antimicrobial drugs are deemed necessary to combat this issue.

## 3. Detection of *Salmonella*

Early and accurate detection of *Salmonella* infection could be a lifesaver, as necessary and effective treatment can be provided to patients, thereby reducing the possibility of selective pressure that can contribute to the emergence of antimicrobial drug resistance [50]. As the technology progresses, there are many new inventions and innovative approaches for *Salmonella* detection that provide fast, accurate, and reliable detection. Figure 1 illustrates the currently available methods for *Salmonella* detection.

### 3.1. Conventional Detection of Salmonella

The clinical diagnosis of typhoid in tropical disease-endemic areas is very difficult and sometimes unreliable. This is due to the difficulties in distinguishing typhoid diagnosis from other co-endemic acute febrile illnesses such as malaria, dengue, leptospirosis, influenza, brucellosis and other systemic infection [50].

The culture method has been the gold standard since the discovery of enteric fever etiological agent in 1880 (as cited in [50]). This traditional isolation method involves enumeration of the targeted bacteria according to their unique morphological and biochemical properties, and it has been standardised by several regulatory agencies [20]. In a clinical diagnostic setting, blood culture is a basic method of detection for *Salmonella*, or else body fluids such as bone marrow, urine, stool, duodenal aspirates, and rose spot extracts can be used as well [51].

To date, the guideline has been standardised by the International Organisation of Standards (ISO) (ISO 6579:2002) for *Salmonella* detection [5]. Through this method, the sample will undergo a pre-enrichment by using buffered peptone water followed by a selective enrichment by using Rappaport Vassialidis soy (RVS) broth and Muller Kauffmann tetrathionate-novobiocin. Finally, the enriched sample will be streaked on a differential medium (e.g., Xylose Lysine Deoxycholate (XLD) and Hoektoen) [5,52]. In general, the proposed guidelines by other regulatory agencies are essentially similar to ISO 6579:2002, which involve four principle stages: non-selective pre-enrichment, selective enrichment, plating on selective isolation agar, and biochemical and serological tests [20,53].

The colonies that appeared on the medium will then proceed for biochemical and serological detection. Until present, a miniaturised form of the biochemical assay was developed to assist in rapid confirmation of isolated organisms from a large number of samples [20]. This system greatly reduces the volume of reagents, media, and apparatus required, as compared to conventional biochemical assay, and produces more results in a short period of time [54,55]. This system comprises a sterilised, disposable microtiter plate, which contains up to 20 specific media or substrates targeting specific microorganisms. Identification of *Salmonella* to species level will be determined after a 16–24 h incubation period at desired temperature. A biochemical identification test kit API 20E and Biolog automated microbial identification system are some examples of commercial biochemical assays available that are still currently being used [56,57].

*Salmonella* serogroup can be assessed through an agglutination test with polyvalent antisera for somatic O antigens. Specific serovar can then be determined by slide agglutination tests with monovalent antisera to specific O antigens and tube agglutination tests with antisera to flagella H antigens [53]. However, there is a critical downside of serotyping identification where the same *Salmonella* serotype could have a different antigenicity. This is due to the loss or modification of surface antigen of the bacterial cells and consequently lowering of the sensitivity of serological tests [58]. If this problem occurs, pulsed-field gel electrophoresis (PFGE) characterisation could be a suitable replacement method [20].

### 3.2. Rapid Salmonella Detection Methods

There is no doubt that the conventional culture method is preferred in many foods’ safety and health diagnostic laboratories as a routine detection due to its ease of use, high sensitivity, reliability, and reasonable operating cost. However, this method requires a minimum of three days to be completed, starting from pre-enrichment to the identification process depending on the biochemical test used [20]. Furthermore, in a certain condition that requires high throughput screening with high number of samples to be analysed, this method might be of huge disadvantage. Some modifications to the conventional method have been made by researchers to reduce labour, time, cost, and it offers much more rapid detection and identification. For instance, the use of fluorogenic and chromogenic growth media (e.g., Rambach agar, SM-ID agar, and BBL CHROMagar *Salmonella*), which shows direct detection, enumeration, and identification on isolation media are some of the good ways to reduce the processing time by one day, compared to the conventional method [59,60,61,62]. Yet, this improvement is still considered lacking in response to *Salmonella* outbreak, bioterrorism, or product recall [20].

To date, several rapid methods with different identification techniques were developed which include immunological-based assays, molecular-based assays, mass spectrometry-based detection, spectroscopy-based detection, optical phenotyping detection, and electrochemical biosensors. From the past two decades, immunological-based, and molecular-based assays are the fastest analytical detection methods developed due to the advancement in molecular sciences. A huge improvement to these methods has been made with the introduction of more specific target antibodies and genes [21]. Vibrational spectroscopy, spectral imaging, and machine vision have gained interest in bacterial detection due to their non-destructive analysis, rapid, and convenient ability to retrieve visual data from specimens and cloud libraries [17]. Another famous method to date is the electrochemical biosensors method, which already received tremendous attention and innovation in research communities in various kinds of applications.

#### 3.2.1. Immunological-Based Assay

Immunological-based assays utilise the specificity of antibodies (monoclonal or polyclonal) for antigen capture that is usually located at the surface of the *Salmonella* cellular membrane [22]. These assays include enzyme-linked immunosorbent assay (ELISA), the latex agglutination method, and immunochromatography assay. Figure 2 summarises the illustration process of immunological-based assays.

Enzyme-linked immunosorbent assay ELISA is the most famous and frequently used labelled immunological-based assay for *Salmonella* detection, where it is considered the gold standard of all immunoassays [4]. It can be used for the detection and quantification of a variety of samples such as antibodies, antigens, hormones, proteins, and glycoproteins [4]. This assay utilises an immobilised anti-*Salmonella* antibody in a solid matrix and, once a specific antigen is bound to the antibody and formed antigen–antibody complex, it triggers colour changes due to enzymatic cleavage of a chromogenic substrate (Figure 2a) [66,67].

Improvement on the ELISA method also has been made with a fluorogenic reporter [68], polymerase chain reaction (PCR) [69] and electrochemiluminescence reporters [70] to improve signal generation. The recent major advancement of ELISA was the manipulation of gold nanoparticles, as chromogenic reporters that greatly improve the sensitivity of detection. This assay provides qualitative results by generating a distinct coloured solution with the presence of analyte. The lowest detection limit was recorded at 1 × 10^−18^ g/mL (attomolar) of prostate-specific antigen (PSA) and HIV-1 capsid antigen p24 in a whole serum sample [71].

The latex agglutination method is one of the simplest *Salmonella* detection techniques, which utilises latex particles coated with antibodies. The presence of *Salmonella* will form a visible latex agglutination as the antigens on the *Salmonella* surface react with the immobilised antibody on latex particles (Figure 2b) [54,72]. Even though this assay is relatively outdated, it can still be used as a confirmatory analysis, since it offers a specific, easy, and reliable technique [73,74,75,76,77].

Immunochromatographic assays also known as lateral flow assays were the basis behind the dipstick assay, lateral flow device (LFD), and lateral flow immunochromatographic assay (LFIA) [6]. This assay consists of four parts namely the sample pad, conjugate pad, reaction membrane and absorbent pad (Figure 2c). The sample pad is the sample receiving region, the conjugate pad is a region where probe hybridisation takes place, the reaction membrane is where the test line and control line for target antigen–antibody or DNA–probe hybridisation interaction is located and absorbent pad served as a waste collection region [65]. These separations of component mixture based on their movement through the reaction membrane are called a chromatographic system [78].

#### 3.2.2. Molecular-Based Assay

Molecular-based assay involves the hybridisation of short oligonucleotide fragment known as a DNA/RNA probe or primer to detect specific targeted DNA/RNA sequences [21]. Specific primer/probe can be isolated from microorganisms or engineered according to their specific target bioreceptor [79]. This method offers high sensitivity and specificity and greatly reduced detection time to only a few hours. They were polymerase chain reaction (PCR), loop-mediated isothermal amplification (LAMP), nucleic acid sequence-based amplification (NASBA), recombinase polymerase amplification (RPA), DNA microarrays, and whole-genome sequencing (WGS). Figure 3 shows the illustration process of a few molecular-based assays based on DNA/RNA amplification.

The polymerase chain reaction (PCR) is considered the gold standard in bacterial identification, as well as in diagnostic application due to its reliability, high accuracy, and very specific detection outcome. PCR utilises the amplification of nucleic acid and has been extensively utilised for *Salmonella* detection and identification for quite some time [7]. PCR works by amplifying or selectively amplifying a unique defined DNA region, by using special molecular ingredients at a specific condition, generating thousands to millions of copies of targeted DNA sequences (Figure 3a) [21,83]. A short turnaround time within 16 h from start until detection has made PCR an excellent tool in a diagnostic application [84,85]. To date, PCR is still being used as a mandatory microbial identification confirmation and PCR technology continues to evolve.

Multiplex PCR (mPCR) is one of the PCR methods developed for multiple detections of microorganisms. This method has been used for *Salmonella* detection and other foodborne pathogens as it allows rapid multiplex detection in various food matrices [82,84,85]. Real-time PCR or known as quantitative PCR (qPCR) is a PCR method that is capable of quantitatively detecting the target sample in real-time [21]. This method can detect the target DNA and bacterial cells as time progresses by using a fluorescent technology known as SYBR green and TaqMan dyes. These dyes will bind on the DNA groove as the double-strand DNA is amplified, which then increases the fluorescent intensity [86,87,88]. qPCR has been widely used for *Salmonella* detection in various food, poultry, and veterinary products [89,90,91]. In the majority of studies, the invasion gene (*invA*) and tetrathionate reductase gene (*ttr*) were the most targeted gene for *Salmonella* identification using the qPCR method [92]. However, as the research progressed, new target genes such as *Salmonella* enterotoxin gene (*stn*) [93], outer membrane porin F gene (*ompF*) [94], hyperinvasive locus A (*hilA*) [95], virulence plasmid gene (*spvC*) [96], and many more have been used, providing high sensitivity and specificity. Multiplex qPCR targeting many genes simultaneously in real-time was also developed to assist in food safety inspection [21].

Even though PCR is considered an excellent choice for reliable and specific microbial identification, there are some limitations and disadvantages to this assay that can contribute to a major problem. In PCR, the presence of walnut components was reported, inhibiting the product amplification [97]. Meanwhile, in qPCR, the presence of fat, black tea and cocoa also inhibit the amplification process [98,99]. PCR is also unable to differentiate between live and dead cells, since it will amplify any targeted sample available regardless of the cellular condition [20]. This lack of discrimination could cause false-positive results, therefore introducing other problems down the line, such as product recall [100]. The PCR process is also laborious, requiring expensive machines (e.g., a thermal cycler, gel electrophoresis, etc.) and trained personnel to conduct and analyse the PCR product [86].

Loop-mediated isothermal amplification (LAMP) is a novel alternative method to amplify DNA aside from PCR. This method can amplify DNA with excellent specificity, rapidity and efficiency under isothermal conditions (constant temperature) [8]. LAMP employs a special DNA polymerase (*Bst*) and a set of four primers that can recognise six distinct target regions of DNA (Figure 3c) [8,101]. This method has several advantages over the traditional PCR, as it requires only a simple heating block, e.g., a water bath or dry bath, to keep the temperature constant during operation, apart from its lesser sensitivity to the PCR inhibitor and higher yield outcome [101,102,103]. Due to these excellent characteristics of LAMP, it has been widely used for *Salmonella* detection [104,105,106]. A multiplex LAMP-based system was also developed to detect multiple bacterial targets. Liu N. et al. developed multiplex-LAMP to detect *Salmonella* spp. and *Vibrio parahaemolyticus* with 100% specificity [107] and EMA-Rti-LAMP assay ethidium bromide monoazide (EMA) treatment to detect and quantify *S*. Enteritidis in real-time [108].

Nucleic acid sequence-based amplification (NASBA) is another isothermal amplification method, which is based on a molecular transcription system targeting exclusively RNA (Figure 3b). [81,109]. This method can selectively detect viable microorganisms in the sample and successfully eliminate the problems that arise from PCR techniques [10]. NASBA was found to be effective and sensitive for *Salmonella* detection [110,111,112].

Recombinase polymerase amplification (RPA) is another new, novel isothermal amplification technique that is capable of amplifying a minimum of 1–10 DNA target copies in less than 20 min [9]. This selective and highly sensitive technique is operated at 37–42 °C, requires minimal sample preparation without prior DNA extraction and purification (Figure 3d). It has been used to amplify ssDNA, dsDNA, RNA, miRNA from diverse samples and organisms [9,113]. Integration of RPA with other detection strategies such as lateral flow [114], real-time fluorescent detection [113], and microfluidic [115] was found to be a success. As soon as their inception, RPA-based methods have been applied in the detection of *Salmonella* in food samples [116,117,118]. Since the success of the isothermal amplification method for *Salmonella* detection, the RPA system is likely the most convenient choice for point of care application due to its short turnaround time, simple implementation, and minimum sample preparation [119].

DNA microarray is an advanced molecular technology that performs a parallel hybridisation of hundreds to thousands of specific and selective DNA probes to their respective target DNA in a single assay [11,27]. This method was initially used to study gene expression analysis [120], but its application had gained widespread expansion for use in comparative genomics, sequence analysis and diagnostics [27]. Reports on microarray-based *S. enterica* detection are available [121,122,123]. Guo D. et al. successfully developed a microarray system that can detect and identify 40 *Salmonella* O serogroups from a simulated food samples [124]. This method is highly specific, as it correctly identifies 98% (*n* = 288) of *Salmonella* strains in a mixed sample with other bacterial species.

Whole-genome sequencing (WGS) or next-generation sequencing (NGS) is another advanced molecular method referring to highly automated and parallelised genome sequencers that are used to sequence the entire genomes of bacterial pathogens [14]. Unlike a DNA microarray that targets certain genes, WGS works by sequencing the entire fragments of bacterial DNA, aligning them into a complete genomic sequence and subsequently compare them in genome sequence databases [23,125]. There are numbers of established genome sequence databases available such as NCBI genome database (https://www.ncbi.nlm.nih.gov/genome/) (accessed on 8 September 2021), CFSAN-FDA (https://github.com/FDA/open.fda.gov) (accessed on 8 September 2021) and other public domains such as GenomTrakr (https://www.fda.gov/food/science-research-food/whole-genome-sequencing-wgs-program) (accessed on 8 September 2021) [14].

With the aid of these databases, WGS can provide detailed information on identified bacterial species including their antimicrobial resistance genes, any mutations to the sequence and even their source-level, such as source of isolation, geographic regions, etc. [14]. Ibrahim G. M. and Morin P. M. in their study successfully predicted *Salmonella* according to their serotypes with a success rate of 89% (*n* = 899) by using WGS and SEQSERO programme as a data analysis tool [12].

#### 3.2.3. Mass Spectrometry-Based Method

The matrix-assisted laser desorption ionisation-time-of-flight mass spectrometry (MALDI-TOF MS) is a recent mass spectrometry technology used for rapid and reliable microorganism identification [126]. This method utilises a protein spectrum profile of a microorganism as a basis of the identification system, which is called a peptide mass fingerprint (PMF) [23,126]. The obtained PMF is compared in an open PMF database, using a scoring algorithm to match the analysed PMF spectrum with reference spectra and consequently identify the microbial species with a high degree of certainty [23].

There are many reports on the use of MALDI-TOF MS for *Salmonella* detection. Mangmee S. et al. develop a MALDI-TOF MS-based method for simultaneous identification of non-typhoidal *Salmonella* (NTS) in the Thai broiler industry. They successfully identified NTS with high accuracy up to species and subspecies level and the method could support faster large-scale screening with efficient cost [13]. Yang S. M. et al. uses MALDI-TOF MS to identify and discriminate three different *Salmonella* serovars, Enteritidis, Typhimurium, and Thompson, that are epidemiologically important in Korea [127].

Liquid chromatography-mass spectrometry (LC-MS) is an alternative mass spectrometry method to MALDI-TOF MS. This method showed promising results in the identification of a closely related bacterial species and has been used for serovar level identification of *Salmonella* ([128] as cited in [14]). This method separates the intact bacterial protein lysates through chromatographic separation and detected with MS [14]. Mc Farland M. A. et al., uses LC-MS coupled with electrospray ionisation (ESI) to identify *Salmonella* serovars (*S*. Typhimurium vs. *S*. Heidelberg) based on their single nucleotide polymorphisms (SNPs) [129]. However, the LC-MS method is slower compared to MALDI-TOF MS analysis [14].

#### 3.2.4. Spectroscopy Method

Raman spectroscopy is a spectroscopy technique based on inelastic scattering of monochromatic light (usually from a laser source). Upon the interaction with the sample, the frequency of photons in monochromatic light changes as they are absorbed by the sample and then reemitted. The shifting frequency of the reemitted photons from the sample is then compared with the original monochromatic frequency to form a Raman effect [23,130]. Surface-enhanced Raman spectroscopy (SERS) is the enhanced version of Raman spectroscopy with a better amplification of electromagnetic fields created by the excitation of localised surface plasmon. SERS can detect samples in a low concentration of analytes with high sensitivity [23]. This technique has been used to rapidly detect foodborne pathogens [15]. Duan N. et al. utilised a SERS-based aptasensor to detect *S*. Typhimurium in a real food sample where they successfully detected the pathogen with a detection limit of 15 CFU/mL [131].

Near-infrared spectroscopy (NIR) is a technique that utilises a specific spectral region called near-infrared (780–2526 nm), where the occurrence of the overtones and combination of vibration response is likely related to the changes in chemical bonds such as O-H, C-H, N-H, C-O, and other organic molecules [23,132]. Since the membrane structure of bacteria consists of combinations of macromolecules that are unique to their respective species, thus the NIR spectrum of each bacterium has a highly specific absorption signature [133]. Pereira J. M. et al. reported fast detection of *S*. Typhimurium in a milk sample by using NIR spectroscopy and evaluated using the chemometric method of partial least squares with discriminant analysis [16]. Their study was able to discriminate *S*. Typhimurium with a good prediction and all samples were correctly classified. Gao X. et al. also reported a compilation studies of foodborne bacteria detection and identification using NIR spectroscopy [133]. This technique provides a non-destructive, fast, and accurate measurement of the target in complex sample matrices.

Hyperspectral imaging (HSI) is an emerging rapid method for bacterial identification. This method utilises the integration of conventional imaging and spectroscopy to provide both spatial and spectral information of a sample [23,134]. Thus, HSI will provide more specific detection and identification compared to a single source of modality. This non-destructive method is usually being used in foodborne pathogen detection and the development of a complete hyperspectral graph of common foodborne pathogens is still under development [135]. Seo Y. N. et al. reported on the development of an HSI method for automated screening of *S*. Enteritidis and *S*. Typhimurium in poultry carcass rinses [136]. They use five different machine learning algorithms to analyse and train the systems and the best performance was achieved by quadratic discriminant analysis (QDA) with a prediction accuracy of 99%.

#### 3.2.5. Optical Phenotyping Using Light Diffraction Technology Method

Microbial identification through light diffraction or a forward light scattering phenomenon is an interesting new way to identify bacteria. This method provides a non-contact and non-destructive measurement as it directly observes the microorganism colonies grown on a culture plate to produce acquisition of scattering images and consequently compare it with reference scatter image libraries of known bacteria [28,137]. BARDOT (Bacterial Rapid Detection using Optical Scattering Technology) and BISLD (Bacteria Identification System by Light Diffraction) are two available methods based on this technique [137,138]. Abdelhaseib M. U. et al. utilised BARDOT the system in their study coupled with a multi-pathogen selective medium to identify *S. enterica*, *E. coli* and *L. monocytogenes* in a single assay [18]. Singh A. K. et al. utilised BARDOT and a laser optical sensor to detect 36 different *Salmonella* serovars in a food sample [139]. Real-time detection in food samples showed 84% detection accuracy in 24 h comparable to those of the USDA Food Safety Inspection Service method, which require ~72 h.

#### 3.2.6. Electrochemical Biosensors

Electrochemical biosensor is an integrated receptor–transducer device that captures a biological signal, processes it through electrochemical means, and translates it into a detectable electrical signal [140]. To date, electrochemical biosensors have received tremendous attention in *Salmonella* detection due to their rapid detection, high specificity and sensitivity and possible on-site testing [23,141,142,143]. A study by Jia F. et al. showed a very sensitive detection of *S*. Typhimurium ATCC 50761 with a limit of detection of 25 CFU/mL in one hour [144]. A similar result was obtained when tested in a spiked chicken sample, which represents good reproducibility. Further details on electrochemical biosensors will be discussed in Section 4.

#### 3.2.7. Advantages and Disadvantages of *Salmonella* Detection Methods

Table 1 summarises the advantages and disadvantages of each discussed *Salmonella* detection method and their examples. The electrochemical biosensor method was found to be the most promising method, as it offers attractive advantages such as low cost per test, user-friendliness, and possible on-site testing. Despite rapid, sensitive, and specific detection, the electrochemical biosensor method struggles with drawbacks such as high early instrument cost (potentiostat), non-standardised sample preparation due to its dependence on the bioreceptor used, and lack of multiplex detection [23,145]. However, recent creation of a low-cost smartphone-controlled potentiostat could solve the expensive potentiostat issue in the near future [146].

Detection sensitivity is one of the important aspects in *Salmonella* detection and it is dependent on the minimum amount of analyte that can be detected. The detection limit represents the lowest concentration of analyte required to give a positive result [154]. The electrochemical biosensor showed the highest sensitivity with the lowest detection limit, and it can detect an analyte down to the femto-molar scale or a small number of cells [155,156]. Meanwhile, in the molecular-based method, the detection limit is expressed in the form of minimum genomic copies for the amplification process. RPA can amplify a minimum of 1–10 DNA targets and PCR requires 32 genome copies in 1 µL of the DNA sample [9,157]. Meanwhile, for other methods, they normally require the initial sample in high concentration. Thus, a pre-enrichment process is one crucial step to increase the analyte concentration.

Cross-reactivity among *Salmonella* serotypes is a common selectivity problem in *Salmonella* identification. This problem not only occurs in immunological-based assay but also in molecular-based identification, which might lead to a false positive result [24,25,26]. This problem can be easily avoided in DNA microarray and whole-genome sequencing (WGS) as a database library of specific serogroups is available [12,124]. Other methods such as MALDI-TOF MS, LC-MS, Raman spectroscopy, NIR spectroscopy, HSI imaging and optical phenotyping usually developed their own specific spectral data, mass fingerprint data or scattering images data that are specific enough to discriminate *Salmonella* serovars [16,127,128,136,139]. In the electrochemical biosensor method, the use of aptamer-based bioreceptor is the best strategy to avoid cross-reactivity problems. In a study conducted by Hyeon J. Y. et al., they successfully developed an RNA aptamer for *S*. Enteritidis with no cross-reactivity to other *Salmonella* serovars [158]. Thus, the use of a highly specific aptamer biorecognition element could prevent cross-reactivity problems. Few studies that utilise aptamer as a biorecognition element showed a good selectivity detection between *Salmonella* serovar [153,159,160].

On the other hand, sample preparation is also a crucial part of the *Salmonella* detection process as the sample might be isolated in variety of forms (solid/liquid). Most of the *Salmonella* detection methods discussed require an input sample in a liquid form except for MALDI-TOF MS, NIR spectroscopy, HSI spectroscopy, and optical phenotyping (BARDOT) as they require bacterial colonies grown on an agar plate [18,127,132,135]. Most of the immunological and molecular-based methods also require sample pre-treatment or pre-enrichment, while other methods such as mass spectrometry, spectroscopy and electrochemical biosensor can be perform in a mixed sample with minimal pre-treatment. These pre-treatment and pre-enrichment processes will increase the overall detection time of sample.

## 4. Biosensors Developed for *Salmonella* Detection

The discovery of a glucose sensor by Clark and Lyons in 1962 was considered the starting point for the development of biosensors in the biomedical field [161]. This technology was then commercialised by the Yellow Springs Instrument Company, by introducing the world first commercial glucose sensor (Model 23A YSI analyser) for the direct measurement of glucose in 1975 [162]. Later, biosensor technology has bloomed with the development of many kinds of biosensor such as cancer biomarker biosensors [163], protein biomarker biosensors [164], and many more.

### 4.1. Biosensors

A biosensor is defined as an analytical device that converts a biological signal or response into a quantifiable and processible signal [165,166]. Generally, a biosensor consists of three main components, namely a biorecognition element, transducer components and the electronic system, e.g., a signal amplifier, processor and electronic display as shown in Figure 4 [167]. The biosensor can detect biological samples in a very specific, fast, and reliable manner utilising a sensing method such as optical, physical changes such as strain or piezo effect due to difference in sample mass or volume, electronic or field-effect transistor (FET) and electrochemical techniques [31]. The generated signal will then be translated into an electrical signal, proportional to the number of analyte–bioreceptor interaction events [31].

Some researchers have also further classified the biorecognition element into analyte and bioreceptor and the electronic system into electronics and display [154]. The analyte is a component of interest or a solution containing a component of interest to be detected such as nucleic acid, proteins, enzymes, antibodies, microorganisms, cellular components, tissues and also drugs [20]. A bioreceptor is a biorecognition element used to detect the presence of the analyte. It could be an enzyme, antibodies, nucleic acid, aptamers, viruses, and cells. A recognition signal will be generated once the target analyte binds to the biorecognition element and the signal can be generated in a variety of ways such as current, charge transfer resistance, potential difference, mass change, absorbance, reflectance, and many more [73,154]. The transducer is an element that converts the biorecognition signals into measurable signals. Electronics is a part that processes the transduced signal through complex electronic circuitry, which then amplifies or transforms the signals into a digital form. Finally, display components will quantify the processed signal generated in a form of a specific graph, numerical data or any format depending on the requirements of the end-user [154].

Biosensors have been broadly classified based on their biorecognition element and transducer [31,168]. These two components are a crucial first step in developing a reliable, sensitive, and robust biosensor combined with good microelectronics systems for data processing and interpretation. The bioreceptor must bind with high affinity to the targeted analyte to ensure high selectivity and sensitivity, and the transducer must be highly sensitive and accurately transform the signals generated for processing [154].

### 4.2. Electrochemical Biosensors

Each type of sensing mechanism has its advantages and disadvantages. Among many kinds of biosensors, electrochemical biosensors have sparked a lot of interest in pathogen detection including *Salmonella* [169]. Electrochemical biosensors offer significant advantages over other biosensors due to their high sensitivity, low cost, versatile detection strategy, automation, miniaturisation potential, sustainability (i.e., low sample volume and minimal solvent use in its development and application) and possible real-time quantification [22,170]. The electrochemical biosensor is also capable of sensing a sample with high turbidity and not being affected by quenching or interference from absorbance and fluorescent compounds as optical-based techniques struggle with [171]. Moreover, the electrochemical biosensor set-up requires relatively simple instrumentation that is low-power compared to surface plasmon resonance techniques that need a light source and receiver, which adds more complexity to the system [171]. This sensing system also can detect a sample in a large linear detection range and a wide variety of solvents and electrolytes compared to a field-effect transistor (FET), which only can detect the target at a certain range of concentration due to a Debye length effect [171,172,173]. Those excellent characteristics are the advantages of electrochemical systems, which make them a reliable sensing system for pathogen detection. A discussion on other transducing elements, such as optical and piezoelectric ones, can be found elsewhere. Figure 5 shows the schematic of typical electrochemical biosensors, including their types of receptors and materials used for electrode surface modification, which will be further discussed in the next section.

Theoretically, the electrochemical biosensor uses electrical means to analyse or examine biochemical reaction (the charge transfer process) that occurs on the surface of the electrode (electrode–solution interface) [166,170]. The electrode itself is one of the crucial components in an electrochemical reaction as its materials, surface modification and dimensions can greatly influence the performance of electrochemical reactions. The working electrode serves as the main region where an electrochemical transduction process takes place, and the counter electrode will complete the circuit by establishing a connection to the working electrode through the analyte as the current is applied. The reference electrode, usually positioned at a distance from the reaction site, serves to maintain a known stable potential [166]. As the reaction takes place, the electrochemical reaction (redox potential changes) can be detected and measured in terms of current, voltage, impedance and capacitance [175,176]. Table 2 summarises the types of electrochemical biosensors based on their electrochemical transduction principle namely potentiometric, amperometric, impedimetric, and voltammetric biosensors.

### 4.3. Bioreceptors Used in Salmonella Biosensors

Bioreceptors or the biorecognition elements are one of the crucial parts in developing *Salmonella* biosensors. The interaction of the bioreceptor and the *Salmonella* will determine the specificity, selectivity, and sensitivity of the developed biosensor [167]. Each type of biorecognition element available has its advantages and disadvantages. Table 3 presents the summary of the pros and cons of each biorecognition element, which are an important consideration for biosensor development.

#### 4.3.1. Antibody–Antigen or Immunosensors

Immunosensors, also known as an antibody–antigen biosensors are a widely utilised analytical tool for *Salmonella* detection especially in dairy and food processing settings [191]. This biosensor works by immobilising a specific anti-*Salmonella* antibody on a surface of a transducer, and the coupling of an antigen to the antibody will trigger an immunochemical reaction, which will be used as a detection signal [192]. Recently, there were quite extensive studies reported on *Salmonella* detection using an immunosensor.

Melo et al. reported on the detection of *S.* Typhimurium in contaminated milk samples via the chronoamperometry method [143]. The polyclonal anti-*Salmonella* antibodies were immobilised on the gold working electrode using carboxymethylated cashew gum (CMCG) film through electrodeposition and the limit of detection (LOD) was observed at 10 CFU mL^−1^ with a detection time of 125 min. Alexandre et al. also reported similar results as Melo in their study for *S.* Typhimurium detection in milk samples with LOD of 10 CFU mL^−1^ and 125 min of detection time [193]. The only difference in this study was that the polyclonal antibodies were immobilised using the cysteine–thiol self-assembly monolayer (SAM) method on a gold screen-printed electrode. A study by Sannigrahi et al. used magnetosome (biogenic nanoparticles synthesised in a magnetotactic bacteria *Magnetospirillum* sp. through biomineralisation) functionalised with anti-*Salmonella* antibody to detect lipopolysaccharide (somatic “O” antigen) of *S.* Typhimurium in food and water samples [194]. Electrochemical impedance spectroscopy (EIS) confirmed the detection of lipopolysaccharide at 0.001–0.1 μg/mL and the LOD of bacteria at 1 × 10^1^ CFU/mL in water and milk samples. A labelled immunoassay was also reported by Bu et al., where they used ferrocene (Fc)-functionalised nanocomposites to amplify the electrochemical signals [195].

#### 4.3.2. Bacteriophage-Based or Phagosensors

A phagosensor is a type of biosensor that uses bacteriophages as their detection vector. Bacteriophages, also known as phages, are viruses that strictly infect and replicate inside bacterial cells [196]. The nature of bacteriophages that only infect a very specific, single-cell host species, and sometimes even specific strains within the species, is an excellent characteristic to employ as a bioreceptor [196]. This strategy is similar to antigen–antibody binding and DNA hybridisation, which provide an exclusive, specific binding of the target [19].

Reports on electrochemical phagosensors for *Salmonella* detection are not widely available as compared to immunosensor and DNA sensors as they are opting for different detection methods. A study by Vinay M. et al. successfully developed a prototype of a phage-based fluorescent biosensor to detect enteric bacteria such as *E. coli* and *S.* Typhimurium in water samples [197]. Their limit of detection was 10 bacteria per mL of solution without concentrating or enriching the sample. Moreover, their prototype is also robust as it can detect target bacteria in seawater sample. Another rapid, sensitive, and direct detection of *S.* Typhimurium based on a wireless magnetoelastic (ME) biosensor has been reported and tested on eggshell samples [198]. This biosensor utilises the E2 phage as the biomolecular recognition element that selectively binds with *S.* Typhimurium and their limit of detection was found at 1.6 × 10^2^ CFU/cm^2^ in a humidity-controlled chamber.

#### 4.3.3. Antimicrobial Peptide-Based Biosensor (AMPs)

Antimicrobial peptides (AMPs) is a short fragment of peptides consisting of around 12 to 15 amino acid residues [199]. AMPs can be found as an innate immune system of living organisms, which provide defence mechanisms against invading pathogens [200]. Generally, AMPs are amphipathic and cationic and they can bind to the bacterial cell membrane via electrostatic and hydrophobic interaction [201]. Their excellent properties such as being intrinsically stable in harsh conditions, their possibility of being produced synthetically in a large quantity, and ease of modification with low production costs, make them suitable candidates as a bioreceptor [168,202].

The number of recent reports for AMP biosensors for *Salmonella* detection is also limited. For example, novel AMPs known as nisin were used in an electrochemical impedance biosensor to detect pathogenic and non-pathogenic *Salmonella* spp. in milk samples. The lowest limit of detection (LOD) was recorded at 1.5 × 10^1^ CFU/mL [142]. Another study by Mannor et al. reported that immobilisation of semi selective AMPs called magainine I on micro capacitive electrode arrays showed a good recognition capability towards *E. coli* and *Salmonella* [203].

#### 4.3.4. Nucleic Acid-Based Sensors or Genosensors

Genosensors also known as DNA biosensors utilise genetic materials such as DNA and RNA sequences as the biorecognition element. Similar to immunosensors and phagosensor, this approach utilises the complementary hybridisation signal of the DNA probe with the target nucleic acid of pathogens, which then translates into a specific transduction signal [204]. In this method, the majority of the target DNA/RNA isolated or extracted from microorganisms undergoes denaturation and is then exposed to the DNA/RNA probe. The hybridisation of probe and target occurs at the sensor surface, which then triggers signal generation [205].

A study by Das Ritu et al. on a *Salmonella* genosensor showed promising results. Biosensor detection response was found to be linear with the target sequence concentrations ranging from 1.0 × 10^−11^ to 0.5 × 10^−8^ M and the lowest detection limit found at 50 (±2.1) pM. This study employed the *Salmonella* Vi gene as a molecular marker. The DNA probe was immobilised on a gold nanoparticle (AuNP)-modified SPE and the signal was monitored using the differential pulse voltammetry (DPV) method [206].

#### 4.3.5. Aptamers as a New Bioreceptor for *Salmonella* Biosensors

Aptamer-based biosensors or aptasensors can be considered as a new technology in biosensors. The aptamer is a single-stranded nucleic acid (ssDNA) or RNA known as oligonucleotides, firstly discovered and reported by Gold and Szostak in the early 1990s [207,208]. The first aptasensor was reported by Kenneth A. Davis and colleagues in 1996 with the development of an optical biosensor based on a fluoresceinated DNA ligand for detection of human neutrophil elastase (HNE) antigen [209]. The aptamer can be obtained through a process called Systematic Evolution of Ligands by Exponential Enrichment (SELEX). Through this process, a pool of nucleotides (nucleotides library) will be incubated with the targeted molecules, and a specific random sequence will form, binding with the target. This process will be repeated for many cycles until a sequence with strong binding to the target molecule is found (Figure 6a) [210]. Aptamers can bind with high affinity and specificity to a broad spectrum of molecules such as nucleotides [211], proteins [212], peptides [213], toxins [214], antibiotics [215], and small molecules [216]. The whole cell SELEX is one of the SELEX methods that received much attention for aptamer development for *Salmonella* [217,218,219]. Through this method, an aptamer specific to the characteristics of the target cell, e.g., specific surface protein, that can be developed and enriched in further steps (Figure 6b) [167].

Aptasensors have also been widely utilised for *Salmonella* detection. For instance, a study by Dinshaw et al. utilised an aptamer specific for the outer membrane of *S.* Typhimurium for detection in artificially spiked raw chicken samples. The aptasensor exhibited a low limit of detection of 10^1^ CFU mL^−1^ [220]. Ma et al. also reported a *Salmonella* biosensor using a *Salmonella*-specific recognition aptamer with a low detection limit of 3 CFU mL^−1^ [221]. Other studies reported a duplex detection of pathogenic microorganisms using an evanescent wave dual-colour fluorescence aptasensor and a fibre nanoprobe. Two fluorescence labelled aptasensors, namely, Cy3-apt-E and Cy5.5-apt-S, for the detection of *E. coli* O157:H7 and *S*. Typhimurium, respectively, were able to perform detection in less than 35 min. The limits of detection were recorded at 340 CFU/mL for *E. coli* O157:H7 and 180 CFU/mL for *S*. Typhimurium [222]. Bagheryan Z. et al. developed a label-free impedimetric aptasensor by using diazonium-modified screen-printed carbon electrodes for the detection of *S*. Typhimurium in food samples [141]. They successfully detect the *Salmonella* with a limit of detection of 6 CFU mL^−1^. Figure 7 illustrates the overview of their aptasensor preparation method and the EIS results obtained. To date, the aptamer biosensor is in the limelight due to its prominent advantages compared to other bioreceptors. Table 4 demonstrates the number of available publications regarding electrochemical aptasensors targeting *Salmonella* sp. from 2010 to 2021.

## 5. Miniaturisation of Electrochemical Biosensors

Sensor miniaturisation could promise an excellent return in terms of possible large-scale production, low production cost, low sample volume analysis, low power analysis, reduced weight, and on-site testing and monitoring on a real-world sample, which is hardly to be achieved with bulky conventional electrodes [226]. Screen-printed electrodes (SPEs) are one of the miniaturised electrode technologies widely utilised in the development of commercial sensors nowadays. They have been used for the detection of heavy metals [227], toxins [228], antibiotics [229] as well as microorganisms [230]. Aside from portability, SPEs are also excellent in avoiding some of the common problems faced by the traditional solid electrode such as memory effects and tedious cleaning process [231]. Figure 8 shows the differences between the conventional electrode platform compared to SPE. The conventional electrodes (e.g., glassy carbon, graphite, etc.) require the presence of external counter electrode (CE) and reference (RE), rigorous polishing and cleaning process prior to use and higher sample volume. Meanwhile, SPE is ready to use without a polishing and cleaning process, smaller in size, disposable and requires much lower sample volume [232]. There are several reports available of SPE-based electrochemical biosensors for the detection of *Salmonella* in various samples (Table 5). However, the reports on the use of SPEs with an aptamer as the biorecognition elements are scarce.

SPEs are usually configured based on three-electrode systems, which include a working electrode, counter electrode, and reference electrode similar to those conventional electrodes. They are printed on an inert, non-conductive substrate such as plastics, ceramic, or a printed circuit board (PCB). The nature of SPE development through layer by layer deposition of a conductive ink using a screen or mesh system on a solid substrate provides wide opportunity and flexibility of using different materials of choice based on the application of interest [226]. Moreover, the introduction of binders, mediators or any conductive components that modify the electrode surface such as polymers or electroactive metals could enhance the sensor efficiency by having better electron transfer kinetics compared to the unmodified electrode [233,234].

However, the miniaturisation process of the sensor does not always translate into good results. Some problems related to the miniaturisation of the sensor have been addressed by Andreas B. Dahlin in his finding [239]. These include the complicated fabrication process as they require more sophisticated and expensive machines to reach precision up to a micro or nano-scale level, are typically more hassle to handle due to their miniaturised size, and produce reduction in measurement precision, which lead to a poor detection limit as well as noise and stability issues [239]. Nevertheless, plasmonic nanoparticle sensors could be one important exception as they are relatively simple to produce and measure on [240]. The development of an open-source format in the form of printed electrodes, circuit boards, electric circuits, components, microcontrollers, software, etc. has made this miniaturised sensor system accessible to the ground as they offer lower production costs [232].

### 5.1. Nanomaterials as the Surface Modifier of Electrochemical Biosensors

In the construction of a sensor, the types of material used, their formula and compositions directly affect the capability and performance of the sensor. The number of particles loaded to build the electrode itself strongly influence the electron transfer process, which translates into a better performance [226]. However, the maximum threshold of the sensor performance is restricted to the surface area of the working electrode itself. This is the area where the miniaturised biosensor is lacking, as the surface area for the biomolecule immobilisation is limited.

To date, the utilisation of nanomaterials as surface modification materials has become eminent. These nanomaterials not only provide a large surface area for biomolecule attachment but also have excellent electrical conductivity, electron transfer and biocompatibility with capture biomolecules [241,242]. Aside from immobilisation, various strategies can also be implemented by using these nanoparticles as an electrochemical label [185]. In general, these nanomaterials can be classified into two groups, which are carbon-based nanomaterials and non-carbon nanomaterials [242].

#### 5.1.1. Carbon-Based Nanomaterials

Graphene and carbon nanotubes are examples of carbon-based nanomaterials. These types of nanomaterials are famous in the research setting and widely applied in the industrial field [243]. Graphene is a two-dimensional (2D) hexagonal pattern of carbon atoms with one single atomic layer, which is densely organised in a regular sp^2^ orbital hybridisation and serves as a basic structure of other sp^2^ carbon-bonded materials such as fullerenes and carbon nanotubes [244,245]. Since its discovery in 2004 [246], graphene-based research has been tremendously popular in a wide variety of research fields with substantial publication beginning in 2013, including the biosensor field [245,247].

Graphene in a 2D hexagonal lattice form has received tremendous attention in sensor research [248]. Its excellent physicochemical characteristics such as large surface area, exceptional electron transfer, high thermal conductivity, good mechanical stability, flexibility and biocompatibility have made graphene a preferable choice in biosensor development compared to other carbon materials [249]. However, the use of the hydrophilic solution as the analyte target of the biosensor (e.g., buffers and fluid samples) has exposed the downside of graphene as graphene itself is hydrophobic. However, this problem can easily encountered by the functionalisation of graphene with hydrophilic functional groups such as a carboxyl group (-COOH) or hydroxyl group (-OH) to obtain a graphitic structure called graphene oxide (GO) [242]. This process can be performed by oxidative stripping of graphite [250,251]. To remove the oxygen-rich functional group on GO, a simple heat or chemical treatment can be conducted to obtain another derivative of graphene called reduced graphene oxide (rGO) [251]. Both GO and rGO retained the basic structure of graphene with modification only applied on the surface and at the edge structure of graphene, but their excellent physicochemical characteristics remain unchanged [245]. To date, there are many biosensor studies reporting on the use of graphene-based nanomaterials for *Salmonella* detection (Table 6). However, since graphene is still considered a new material of interest in the biosensor field, the reliability and reproducibility for high-performance analysis and real-world performance are still under development.

Carbon nanotubes (CNTs) are a carbon structure with an electron orbital hybridisation type sp^2^ between adjacent carbon atoms such as graphene. Unlike sheet-structured graphene, the CNTs’ structure comprises a distinct tube-like hollow cylindrical carbon nanostructure, which is made up of rolled-up graphene sheets [244,252]. This structure grants CNTs a strong mechanical structure, large surface area and excellent electrical conductivity [248]. Most of the carbon nanotubes’ physical properties are derived from graphene [244]. There are two types of CNTs, namely single-walled carbon nanotubes (SWCNTs) and multi-walled carbon nanotubes (MWCNTs).

As the name suggests, SWCNTs are made up of a single cylindrical shape of carbon nanostructure while MWCNTs are composed of multilayer concentric single-walled graphene cylinders supported by Van der Waals forces [242]. MWCNTs have a few advantages over SWCNTs such as high purity, easy bulk synthesis, catalyst-free synthesis, being less prone to defects, have more accumulation in the CNTs’ body (provides more surface area) and having a rigid structure. Meanwhile, SWCNTs offer less purity, are difficult to synthesis in bulk, require a catalyst for synthesis, are prone to defects during functionalisation, have less accumulation of the CNTs’ body and can easily bend and be twisted [244]. Both CNTs have been used in biosensor technology to enhance the electrical properties. Their huge surface area could provide a better immobilisation base for the bioreceptor, improved electrical conductivity and translated into a better signal response [253].

#### 5.1.2. Non-Carbon Nanomaterials

Non-carbon nanomaterials are nanomaterials that originated from sources other than carbon such as metals, silica, polymers and many more. They have been employed as alternative supporting materials of the electrode to improve the electrochemical performance of biosensors. Examples of non-carbon nanomaterials are metallic nanoparticles, silica nanoparticles, nanowire, indium tin oxide (ITO) as well as organic polymers and they have become more popular options in biosensor research [242]. Table 6 shows some of the *Salmonella* biosensors that utilise non-carbon nanoparticles in their systems.

Gold nanoparticles (AuNPs) are the most famous metallic nanoparticles studied in the biosensor field [269]. They comprise thousands of atoms, which can be electrochemically oxidised or reduced and essentially serve as an electrochemical mediator to improve electron transfer [270]. They offer a large surface to volume ratio due to that nanoscale size, unique electron transfer ability, and great biocompatibility [271]. AuNPs also can be handily conjugated with many biomolecules without affecting their respective biochemical activities [272]. Thiolation of respective biomolecules, e.g., oligonucleotides (DNA-AuNPs) is probably the most simple and best way to attach biomolecules to AgNPs through a self-assembled monolayer (SAM) [273]. In the electrochemical DNA sensor field, AuNPs can be utilised through many kinds of strategies such as a modified DNA probe with AuNPs for signal enhancement, DNA hybridisation detection through AuNP labels, surface modification of electrode with AuNPs to increase DNA probe adsorption, and silver deposition on AuNPs [270]. There are other metallic nanoparticles such as silver, copper, and platinum nanoparticles, but the research on their function as *Salmonella* biosensors are quite scarce. Iron oxide (Fe_3_O_4_) magnetic nanoparticles are also an excellent nanoparticle in biosensor study. It has been used in various research fields such as biotechnology, pharmacology, drug delivery and cell separation. This is due to their excellent superparamagnetic property, biocompatibility, fast electron transfer, low toxicity, and good catalytic action [274,275].

Mesoporous silica nanoparticles (MSN) have received interest in biosensors due to their high load capacity, simple preparation and easy manipulation of morphology, size and pore diameter [276]. An excellent electron transfer of MSN could be achieved through proper manipulation, design and tailoring of the nanostructure [277]. In biosensors, MSN are usually used as a capture agent or controlled release process of biomolecules or redox probes by external stimuli [278].

Indium tin oxide (ITO) is usually employed as an electrode due to its great electrical conductivity, cost-effective nature and special optoelectronic characteristics and transmittance [279,280]. ITO contains hydroxyl groups (-OH) on its surface, which can be further functionalised with a wide variety of chemical compounds to capture biomolecules [242]. ITO can be deposited as a thin film on the electrode and its glass-like properties provide an effective immobilisation of biomolecules through surface modification. However, ITO has a slow electron transfer rate compared to a metallic and carbon-based electrodes [242,248].

Nanowires are a cylindrical nanostructure quite similar to those of carbon nanotubes, but with a lower length to diameter ratio. Usually, the aspect ratios of nanowires are in the range of thousands while nanotubes are far beyond that [281]. Nanowires have an excellent surface to volume ratio, electron transfer properties and improved charge carrier motions compared to bulky wires [281,282]. They can be synthesised from metals (e.g., Cu, Ni, Pt, Au, etc.), metal oxides (e.g., Fe_2_O_3_, ZnO, SnO_2_, etc.) and semiconductors (e.g., Si, InP, GaN, etc.), where each material will directly influence their electrical conductivity [248]. For instance, silver nanowires were found to have extraordinary electrical properties such as rapid response, excellent electrocatalytic behaviour and reproducibility, which then was used as a component in electrochemical immunosensors [277].

The conductive polymer is a class of organic polymer, which has characteristics similar to some inorganic semiconductors and metals (e.g., good electrical conductor) while retaining polymer properties such as flexibility, ease of synthesis and processing, low cost and possible miniaturisation [283]. It can be printed on diverse solid supports and has been used in transistor and electrode fabrication [242]. For example, the conductive organic polymer has been used in glucose oxidase sensors by incorporating organic thin-film transistors (OTFT) into the sensor [284]. To date, conductive organic polymers are usually used as a synergy combination nanocomposite with metal nanoparticles or graphene derivatives. This combination has received considerable attention due to the excellence in electrochemical properties, as well as improved catalytic stability [285].

### 5.2. Lab-on-Chip Platforms for Rapid Detection of Salmonella

A lab-on-chip (LoC) is a system that brings all the analytical assays conducted on a laboratory scale into a single, miniaturised, and autonomous device. This system works by integrating microfluidic technology coupled with biochemical or chemical processes that serially work to conduct the analysis [286]. Some diagnostic applications that successfully utilise the LoC concept are glucose and HIV detection systems [287,288].

LoC systems have also been implemented for pathogen detection including *Salmonella*. For example, Tsougeni et al. developed a LoC platform based on an oxygen plasma nanotextured polymeric chip containing bacteria immunoaffinity capturing chamber, chemical lysis, DNA isothermal amplification and a label-free Surface Acoustic Wave (SAW) biosensor. This system successfully detects the bacteria in less than 4.5 h [289]. Another LoC system integrated with loop-mediated isothermal amplification (LAMP) showed a promising detection result within 1.5 h with a low detection limit of 14 CFU mL^−1^ [290]. The Verebeef^TM^ Detection Kit is one of the commercial detection kits that incorporate multiplex PCR and microarray technology on the LoC system for the detection of multiple pathogenic microorganisms including *Salmonella* species. This kit has been extensively tested on raw beef trim samples, and it was able to successfully detect each pathogen without any false-positive or false-negative detection [291].

Even though there are only a few numbers of LoC systems have been reported for *Salmonella* detection, the likelihood of finding the LoC system integrated with an electrochemical system, especially the electrochemical aptasensor system, is scarce. This might be due to the aptamer technology still being at its early stage in miniaturised biosensor technology and probably progressing well in the near future. The integration of LoC systems with a miniaturised electrochemical aptasensor could be a huge leap on *Salmonella* detection, as this system can provide a very specific, rapid, sensitive, and sustainable microorganism detection.

## 6. Conclusions and Future Impact of *Salmonella* Biosensors

The research on pathogen detection, especially on the *Salmonella enterica* species, is still progressing from year to year, emphasising the importance of a rapid and precise detection to prevent any casualty and fatality to humans. Even though the world statistics of *Salmonella* infection are gradually declining, provident preparation must be made to prevent any potential outbreak. In this review, we have presented the progression of *Salmonella* detection technology from the conventional culture-based method to an advanced electrochemical biosensing system. The conventional culture method is still being used nowadays, especially in clinical settings where the target sample (e.g., blood, plasma, stool, etc.) is sometimes difficult to process with the latest sensing technology or the samples might need to be clinically diagnosed in detail, such as for their antimicrobial resistance properties, etc. Rapid detection methods based on immunological assay, molecular assay, mass spectrometry, spectroscopy, optical phenotyping are also reliable and they have their specific niche in the *Salmonella* detection system. Biosensors are undoubtedly one of the best *Salmonella* detection systems available, offering versatile applications in various fields, such as medical diagnostic, food safety, environmental monitoring, drug delivery, and many more. The bioreceptor is one of the important aspects of the biosensor to ensure specific and selective detection of the target. Amongst the discussed bioreceptors (i.e., antibody–antigen, bacteriophage, AMPs, nucleic acids, and aptamer), the aptamer was found to be the most reliable bioreceptor due to its outstanding advantages compared to that of the others. The integration of nanomaterials to the already sensitive electrochemical sensing technology can further enhance the *Salmonella* detection limit down to the femtomolar concentration, as well as the detection of individual *Salmonella* cells. Theoretically, these combinations could be a plausible move for the development of *Salmonella* point of care detection devices (in a form of a lab-on-chip device) that have rapid, highly specific, selective, and sensitive detection, are small enough for on-site application (miniaturised form), are potentially easy and cheap to build, and are user friendly. Furthermore, the development of big data analytics and artificial intelligence (AI) technology could further enhance *Salmonella* biosensor technology to reach the next milestone.

## Figures and Tables

**Figure 1 biosensors-11-00346-f001:**
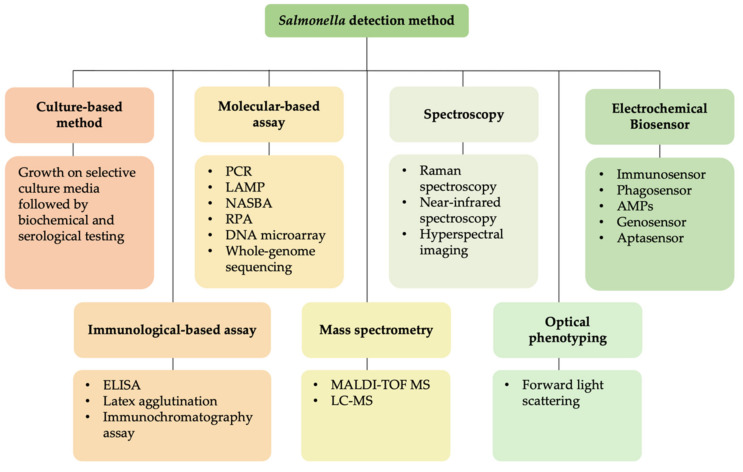
Available methods for *Salmonella* detection (adapted from ref. [14,20,21,23]).

**Figure 2 biosensors-11-00346-f002:**
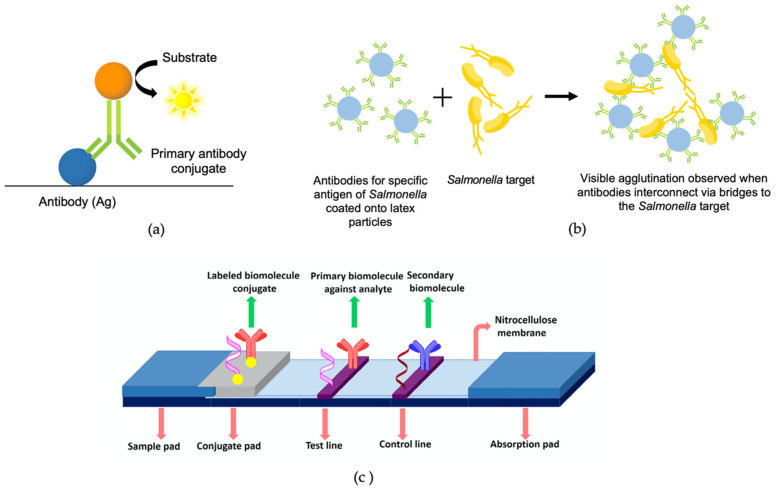
Illustration of a rapid *Salmonella* detection method based on immunological methods: (**a**) schematic diagram of direct enzyme-linked immunosorbent assay (ELISA) that generates a coloured signal with the presence of the target (adapted from ref. [63]); (**b**) illustration of the latex agglutination method that formed a visible clump in the presence of *Salmonella* (adapted from ref. [64]); and (**c**) the basic structure of a lateral flow assay (reproduced from ref. [65] with permission from Elsevier).

**Figure 3 biosensors-11-00346-f003:**
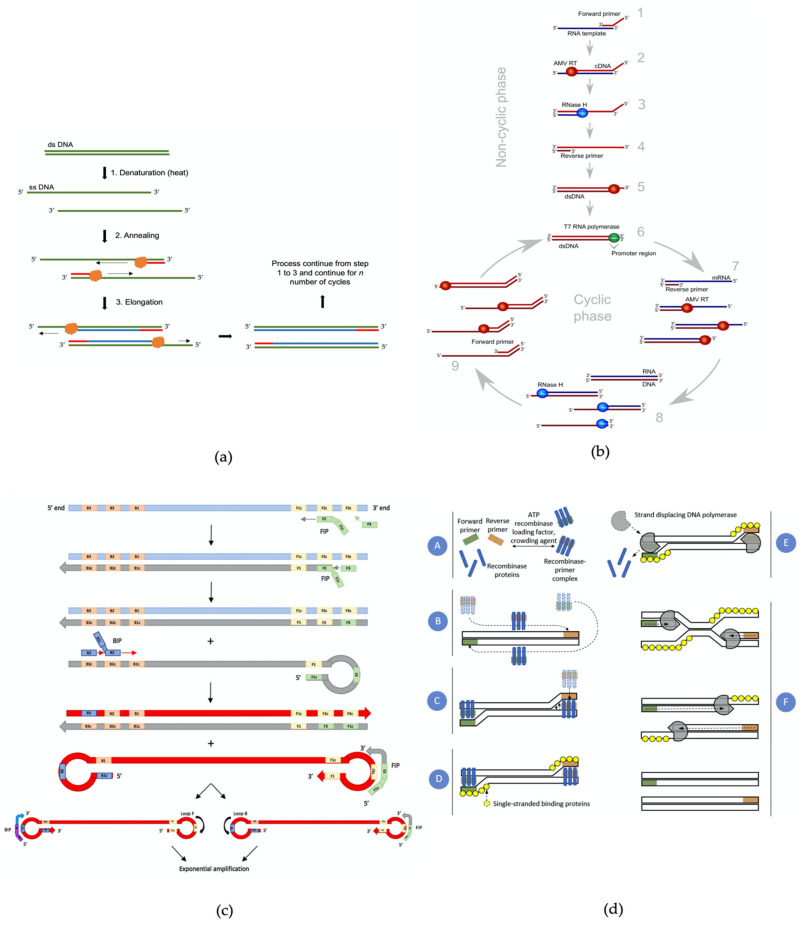
Schematic illustration of a molecular detection method based on an amplification technique: (**a**) the polymerase chain reaction (PCR) amplification process (adapted from ref. [80]); (**b**) the nucleic acid sequence-based amplification (NASBA) process (reproduced from ref. [81] with permission from Elsevier); (**c**) the loop-mediated isothermal amplification (LAMP) process (reproduced from ref. [82] with permission from John Wiley and Sons); and (**d**) Recombinase polymerase amplification (RPA) flow (reproduced from ref. [9] with permission from Elsevier).

**Figure 4 biosensors-11-00346-f004:**
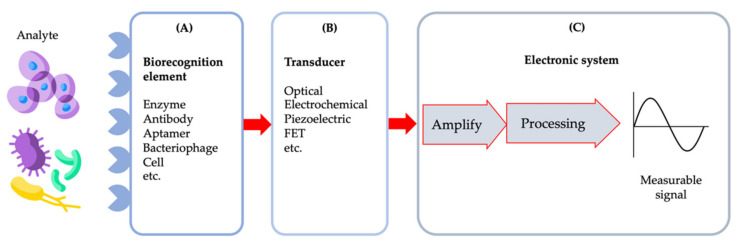
Schematic diagram of three main components of a biosensor and examples (adapted from ref. [167]): (**A**) biorecognition elements that interact with the analytes such as cells, tissues, antigens, nucleic acids and many more; (**B**) transducer elements that convert the analyte–bioreceptor interaction into a quantifiable signal; and (**C**) electronic systems amplify and process the signal from the transducer and display the output as digital data.

**Figure 5 biosensors-11-00346-f005:**
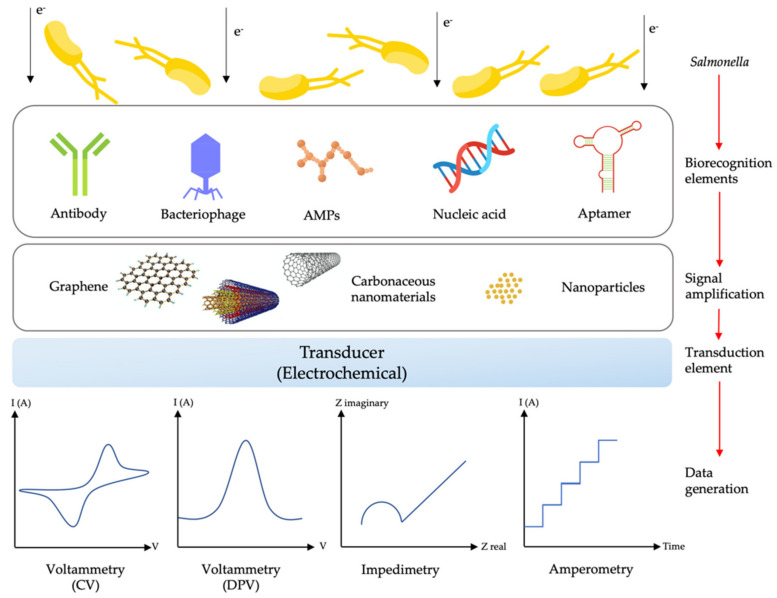
Schematic illustration of a typical electrochemical biosensor with different types of biorecognition elements, materials for signal amplification and electrochemical transducing techniques for *Salmonella* detection (adapted from ref. [22] and the graphene scheme from ref. [174]).

**Figure 6 biosensors-11-00346-f006:**
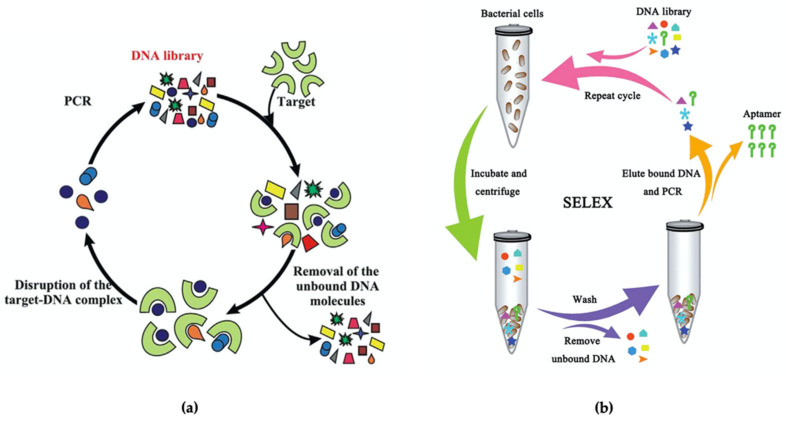
(**a**) Schematic diagram of the SELEX method. (**b**) Whole-cell SELEX method for aptamer selection against live bacterial cells. (Reprinted with permission from ref. [167]).

**Figure 7 biosensors-11-00346-f007:**
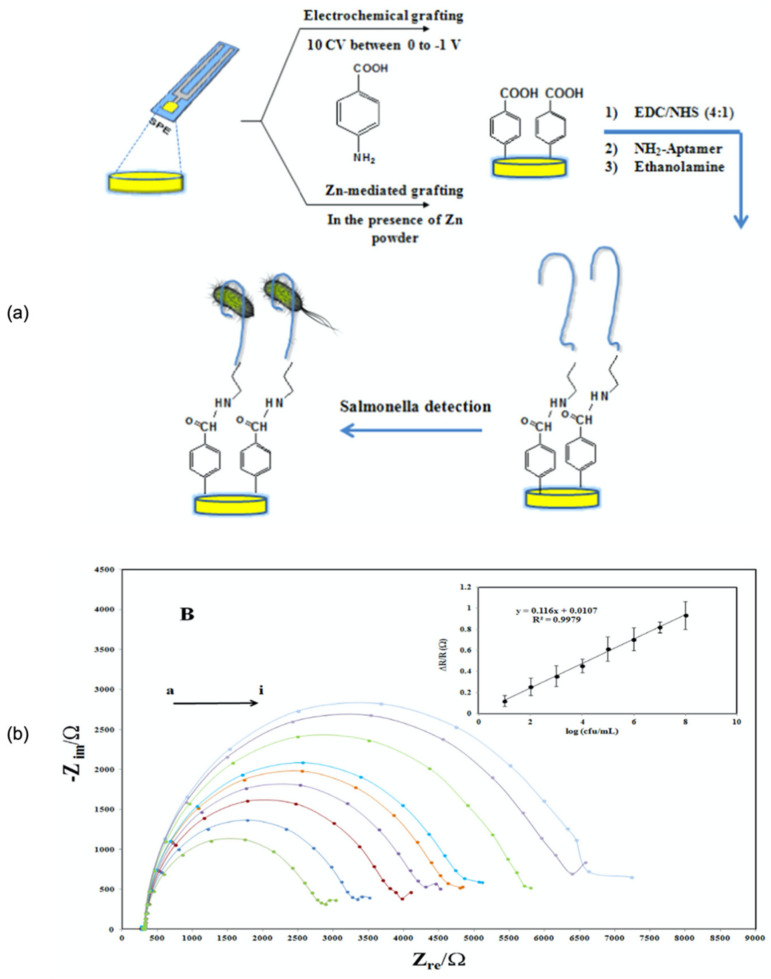
(**a**) The overview of the *Salmonella* aptasensor preparation utilising a screen-printed electrode (SPE); and (**b**) EIS results of aptasensor-based charge transfer resistance (*R*ct) when incubated with different concentrations of *S*. Typhimurium. This figure has been reproduced from ref. [141] with permission from Elsevier.

**Figure 8 biosensors-11-00346-f008:**
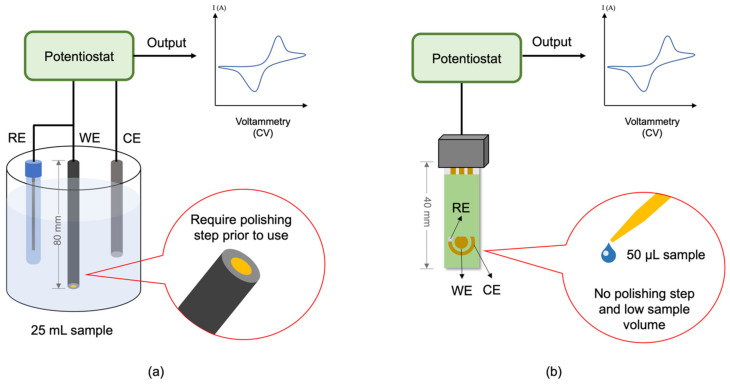
Schematic overview of a classical electrochemical sensing process using a conventional electrode (**a**) and SPE (**b**) adapted from ref. [232].

**Table 1 biosensors-11-00346-t001:** Overview of *Salmonella* identification method used in the literature including their advantages, disadvantages, and examples.

Detection Method	Advantages	Disadvantages	Example	^a^ Time to Results	Detection Limit	Ref.
			Input Sample			
**Conventional method**						
Microbiological culture method	Sensitivity, selectivity with chromogenic media, low cost	Labour intensive, lengthy analysis, requires a sterile environment, no on-site testing, VBNC	Food samples **	Several days (3 to 7 days)	-	[52]
**Immunological-based method**						
ELISA	High specificity, rapid, user friendly, high throughput, qualitative and quantitative	Batch to batch variation, low sensitivity, antigen cross-reaction, difficult synthesis of target antibody	Chicken meat **	2–3 days (detection in few hours)	-	[147]
Latex agglutination method	High specificity, user friendly, high throughput, low cost, rapid detection	Qualitative, susceptible to false negative results	Blood culture broth **	2–3 days (detection in few hours)	-	[148]
Immunochromatographic assay	Lightweight, disposable, on-site testing, low detection limit compared to other immunological method	Antibody instability	Water sample **	2–3 days (detection in 5–7 min)	10^4^ CFU mL^−1^	[149]
**Molecular-based method**						
PCR	High sensitivity and specificity, reliability, multiplex detection (mPCR), real-time detection (qPCR)	PCR inhibitor, unable to distinguish between live and dead cells, expensive equipment, requires well-trained personnel	Seafood sample **	2 days (detection in few hours)	-	[150]
LAMP	High specificity, real-time detection, requires less equipment due to isothermal properties, less sensitive to PCR inhibitor, low detection limit	Complex sample preparation, indirect detection method	Pork product and carcass **	2 days (detection in 90 min)	10^1^ CFU mL^−1^	[105]
NASBA	High specificity, rapid, able to detect only viable cell, low detection limit	RNA is more labile than DNA	Beef, pork, and milk *	<24 h (detection in <2 h)	<10^1^ CFU mL^−1^	[110]
RPA	High specificity, rapid, minimal sample preparation, low detection limit	-	Milk *	<12 h (detection in 15 min)	50 CFU mL^−1^	[118]
DNA microarray	Easy to operate, online database for analysis and comparison	Expensive equipment, requires well-trained personnel for analysis	Spiked tomato sample **	2 days	-	[124]
Whole genome sequencing	Highly automated, wide variety of databases publicly available, able to detect genotypic	Expensive equipment, requires well-trained personnel	Lettuce *	3–4 days	-	[151]
	characteristics of bacteria such as antimicrobial susceptibility or virulent profile					
**Mass spectrometry method**						
MALDI-TOF MS	High sensitivity and selectivity, high throughput, rapid, low cost per testing, non-destructive, can be used in a complex sample	High early instrument cost, database limitations, no on-site testing, requires well-trained personnel	Blood culture **	1–2 days (detection in <5 min)	-	[152]
LC-MS	High sensitivity and selectivity,	Expensive equipment, requires well-trained personnel,	Pure culture **	2 days (detection in few hours)	-	[129]
**Spectroscopy method**						
Raman spectroscopy	High sensitivity, high specificity, non-destructive, culture independent, multiplex detection, easy device handling	Fluorescent background, no on-site testing, expensive equipment,	Pure culture **	2 days (detection in few hours)	-	[131]
Near-infrared spectroscopy (NIR)	High sensitivity, non-destructive, real-time detection, real-time detection	Signal saturation due to water content, no on-site testing, expensive equipment, requires well-trained personnel,	Milk **	2 days (detection in few hours)	-	[16]
Hyperspectral imaging (HSI)	High specificity and selectivity, non-destructive, real-time detection	High detection limit, no on-site testing, expensive equipment, requires well-trained personnel,	Chicken carcass rinse **	2 days (detection in few hours)	-	[136]
**Optical phenotyping method**						
Light diffraction/forward light scattering	High specificity and selectivity, non-destructive, real-time detection	database limitations, no on-site testing, expensive equipment, requires well-trained personnel,	Peanuts, spinach, chicken carcass, pork, and turkey samples **	2 days (detection in few hours)	-	[139]
**Biosensor**						
Electrochemical biosensor	High sensitivity and specificity, rapid, high throughput, user friendly, low cost per testing, real-time detection, low detection limit, on-site testing	High early instrument cost, sample preparation depends on bioreceptor	Raw chicken sample *	4 h (detection in 5 min.)	10^1^ CFUmL^−1^	[153]

^a^ Time to results is the time taken from initial sample processing until detection results. ** Require a long pre-treatment and enrichment of sample >16 h. * Require simple pre-treatment for few hours.

**Table 2 biosensors-11-00346-t002:** Summary of electrochemical biosensor types according to their transduction principle, their performances and examples of studies conducted.

Types of Electrochemical Biosensor	Working Mechanism	Advantages	Disadvantages	Biosensor Developed	Ref.
Potentiometric	Measure the charge accumulation (potential) on the working electrode due to the interaction between the analyte and bioreceptor relative to the reference electrode under zero or negligible current flow. Usually, an ion-selective electrode and ion-sensitive field-effect transistors are used.	-Miniaturisation potential -Electrode surface area does not affect signal		Aptasensor for *Salmonella* detection using an ion-sensitive electrode (ISE) modified with single-walled carbon nanotubes (SWCNT).	[177]
				Immunosensor for *S*. Typhimurium detection using cadmium and sodium ion-selective electrodes as an indicator and pseudo-reference electrodes.	[178]
				Immunosensor for *S*. Typhimurium detection using a paper strip ion-selective electrode integrated with a filter paper pad as solution reservoir.	[179]
Amperometric	Measure the current produced at the working electrode due to electrochemical oxidation or reduction of electroactive species when a constant potential is applied with respect to the reference electrode. Amperometric biosensor can operate in either two or three electrodes. The current produce is proportional to the analyte concentration present in the solution.	-Suitable for mass production -Sensitive, fast, precise, and provides a linear response compared to potentiometric biosensor	-Poor selectivity -Interference from other electroactive substances	Label-free immunosensor for *S*. Typhimurium detection using the as-grown double wall (DW) carbon nanotube bundles as an electrode and chronoamperometry as transducing method.	[180]
				Immunosensor for *S*. Typhimurium detection using an enzymatic substrate and mediator for response detection	[181]
Impedimetric	Measure the electrical impedance (change in electrical conductance or capacitance) produced at the electrode/electrolyte interface in a constant potential.	-Miniaturisation potential -Fast response	-Signal instability due to the electrode to electrode and probe variations	Aptasensor for *S*. Typhimurium detection using a diazonium-supporting layer SPE in spiked apple juice.	[141]
				Immunosensor for *S*. Typhimurium detection using cetyltrimethyl ammonium bromide (CTAB) functionalised MoS2 nanosheets (CTAB-MoS2-NS) for protein conjugation on a microfluidics ITO-hydrolysed microelectrode.	[182]
				Immunosensor for *S*. Typhi detection using gold nanoparticle (AuNPs)-tagged bacteria via high-affinity antigen–antibody interactions in interdigitated microelectrodes.	[183]
Impedimetric				Aptasensor for *S*. Typhimurium detection using an aptamer-coated gold interdigitated microelectrode for target capture and antibody modified nickel nanowires (NiNWs) for magnetic target separation in the spiked chicken sample.	[184]
Voltametric	Measure the changes in current during the controlled variation of applied potential.	-Highly sensitive measurement -Simultaneous detection of multiple analytes -Less prone to noise		Immunosensor for *S*. Typhimurium LT2(S) detection using magneto-immunoassay and gold nanoparticles (AuNPs) as a label in the skimmed milk sample.	[185]
				Aptasensor for *S*. enterica detection using a pencil graphite electrode decorated with chitosan (Chi)-electrospun carbon nanofibers (CNF)/gold nanoparticles (AuNPs).	[186]
				Immunosensor for *S*. Typhi detection using a disposable microfluidic device (DμFD) based on a carbon electrode array and magnetic gold nanoparticles (AuNPs) as a label.	[187]
				Aptasensor for S. Typhimurium detection using a metal-organic framework–graphene composite of type UiO-67/GR as a base substrate and aptamer–gold nanoparticles–horseradish peroxidase (Apt-AuNP-HRP) conjugate as the signal amplification probe.	[188]

The details on the types of electrochemical transducing mechanisms and their advantages and disadvantages were adapted from ref. [22,31,145,166,168]. Limit of detection (LOD); colony-forming units (CFU); phosphate buffer solution (PBS).

**Table 3 biosensors-11-00346-t003:** Summary of biorecognition elements used in biosensors.

Bioreceptor	Advantages	Disadvantages
Antibody	High affinity and specificity Possible for reusability	Low stability Possibility of batch variation High cost Laborious production
Bacteriophage	Can discriminate live and dead cells Phage structures can be engineered for better affinity, specificity, and robustness Low cost	Potential of bacterial lysis during detection Low capture efficiency when dry
AMPs	High affinity and stability Simple synthesis process Access to modification Low cost	Low specificity
Nucleic acid	High stability Simple synthesis process Access to modification Low detection limit	Laborious production Low specific binding Restricted to DNA target only
Aptamer	High affinity, stability and specify Simple synthesis process Access to modification Low cost Low detection limit	Sensitive to nuclease

Adapted from ref. [168,189,190].

**Table 4 biosensors-11-00346-t004:** Research published on electrochemical aptasensors for the detection of *Salmonella*.

No.	Serotype	Aptamer Target	Sample	Immobilisation Method	Detection Method	Linear Detection Range (CFU mL^−1^)	Limit of Detection (CFU mL^−1^)	Time to Results/Detection Time	Ref.
1	*S. enterica*	-	-	Chitosan (Chi)-electrospun carbon nanofibers (CNF)/gold nanoparticles (GNPs)	DPV	10–10^5^	1.223	-	[186]
2	*S*. Typhimurium	Outer membrane proteins (OMPs)	Food/raw chicken sample	Reduced graphene oxide–carbon nanotubes (rGO-CNT)	DPV	10^1^–10^8^	10^1^	4 h/5 min	[153]
3	*S*. Typhimurium	Outer membrane proteins (OMPs)	Chicken meat	Reduced graphene oxide–titanium dioxide (rGO-TiO2) nanocomposite	DPV	10^1^–10^8^	10^1^	5 h/60 min	[223]
4	*S. enterica*	Vi polysaccharide antigen	sera and urine specimen	Molybdenum disulfide (MoS2) and reduced graphene oxide composite	CV, DPV and SWV	0.1 ng mL^−1^–1000 ng mL^−1^	0.1 ng mL^−1^	3 h/45 min	[224]
6	*S*. Enteritidis *S*. Typhimurium	Outer membrane proteins (OMPs)	-	Multi-walled carbon nanotubes (MWCNTs)	CV and EIS	5.5 × 10^1^–5.5 × 10^6^ 6.7 × 10^1^–6.7 × 10^5^	5.5 × 10^1^ 6.7 × 10^1^	-/10 min	[225]
7	*S*. Typhimurium	Outer membrane proteins (OMPs)	-	Reduced graphene oxide–chitosan (rGO-CHI) composite. Glutaraldehyde as a cross linker	CV and DPV	10^2^–10^6^	10^1^	-	[220]
8	*S*. Typhimurium	Outer membrane proteins (OMPs)	Apple juice	Diazonium supporting layer	EIS	10^1^–10^8^	10^1^	2 h/30 min	[141]
10	*Salmonella* sp. ATCC 50761	-	Spiked fresh chicken	Reduced graphene oxide (rGO) and carboxy-modified multi-walled carbon nanotubes (MWCNTs)	EIS	75–7.5 × 10^5^	25	Few hours/60 min	[144]
11	*S*. Typhimurium	-	Spiked chicken	Gold interdigitated microelectrode	EIS	10^2^–10^6^	80	-/120 min	[184]
12	*S*. Typhimurium	-	Spiked mineral water and milk	Gold nanoparticles (AuNPs)	CV and EIS	20 to 2 × 10^8^	15	-	[156]
13	*S*. Typhimurium	Outer membrane proteins (OMPs)	Spiked food sample	Poly [pyrrole-co-3-carboxyl-pyrrole] copolymer	EIS	10^2^–10^8^	100	2 h/45 min	[159]

DPV: Differential pulse voltammetry, CV: cyclic voltammetry, EIS: electrochemical impedance spectroscopy, SWV: square wave voltammetry. Time to results is the time taken from initial sample processing until detection results.

**Table 5 biosensors-11-00346-t005:** *Salmonella* electrochemical biosensors utilised in the use of SPE.

Working Electrode	Signal Monitoring	Surface Modification	Biomarker	Ref.
Carbon	DPV	Gold nanoparticle (AuNp)	*S*. Typhi Vi gene	[206]
Carbon	CV	Gold nanoparticle (AuNp) and ionic liquid	Antibody	[235]
Gold	CV	Cysteamine	Antibody	[193]
Gold	CV	Carboxymethyldextran	Antibody	[236]
Gold	EIS	Cysteamine	Antibody	[237]
Carbon	CV	Fe_3_O_4_/SiO_2_/AuNPs nanocomposites	Antibody	[238]
Carbon	EIS	Diazonium supporting layer	Aptamer	[141]

**Table 6 biosensors-11-00346-t006:** Integration of nanomaterials in biosensor application for *Salmonella* detection.

Class of Nanomaterials	Types of Nanomaterials	Biorecognition Elements	Transducer Type	Detection Technique	Input Sample/Sensitivity (LoD)	* Analysis Time	Ref
**Carbon-based nanomaterials** Graphene’s derivatives	Laser induced graphene electrode	Immunosensor	Electrochemical	EIS	Chicken broth/13 ± 7 CFU mL^−1^	48 h/22 min	[254]
	rGO–MWCNTs nanocomposite	Aptasensor	Electrochemical	EIS	Spiked fresh chicken/25 CFU mL^−1^	Few hours/60 min	[144]
	rGO–CNTs nanocomposite	Aptasensor	Electrochemical	DPV	Raw chicken sample/1 × 10^1^ CFU mL^−1^	4 h/5 min	[153]
	rGO–chitosan complex	Aptasensor	Electrochemical	CV and DPV	1 × 10^1^ CFU mL^−1^	-	[220]
	rGO–polypyrrole nanocomposite	Genosensor	Electrochemical	DPV	8.07 × 10^1^ CFU mL^−1^	-/60 min	[255]
	Carboxylated GO decorated with Fe_3_O_4_ NPs	Genosensor	Electrochemical	DPV	3.16 × 10^−18^ M	-	[256]
	Graphene oxide-modified SPE	Phagosensor	Electrochemical	EIS	1 × 10^−1^ CFU mL^−1^	-/4 min	[257]
Carbon nanotubes (CNTs)	MWCNTs	Aptasensor	Electrochemical	EIS	5.5 × 10^1^ CFU mL^−1^ for *S*. Enteritidis; 6.7 × 10^1^ CFU mL^−1^ for *S*. Typhimurium	-/10 min	[225]
	SWCNTs	Genosensor	Electrochemical	EIS	1 × 10^−9^ mol L^−1^	-	[258]
	SWCNTs	Aptasensor	Optical	Chemiluminescent	City water sample/1 × 10^3^ CFU mL^−1^	48 h/-	[259]
**Non-carbon nanomaterials** Metallic nanoparticles	AuNPs	Immunosensor	Electrochemical	DPV	Skimmed milk/143 cells mL^−1^	-/90 min	[185]
	AuNPs	Oligonucleotides	Optical	Colorimetric	Chicken meat and blueberry/<10 CFU mL^−1^	48 h/~35 min	[260]
	AuNPs–chitosan composite	Immunosensor	Electrochemical	DPV	Tap water and milk/5 CFU mL^−1^	-/240 min	[261]
	silver nanoclusters (AgNCs)	Genosensor	Electrochemical	DPV	0.162 fM	-/80 min	[262]
Silica nanoparticle	Mesoporous silica nanoparticles capped with Zinc oxide (ZnO)	Immunosensor	Optical	Colorimetric Fluorescent	Chicken meat sample/63 CFU mL^−1^ 40 CFU mL^−1^	-/90 min	[263]
Silica nanoparticle	Mesoporous silica Nanoparticles–GO–cobalt nanocomposite	Aptasensor	Electrochemical	EIS	~1 × 10^1^ CFU mL^−1^	-	[264]
Indium tin oxide (ITO)	3D nanostructured indium–tin oxide (ITO)	Lipopolysaccharide (LPS)	Electrochemical	EIS	2–3 ng mL^−1^	-	[265]
	Gold modified ITO electrode	Aptasensor	Electrochemical	DPV	10 fM	-	[155]
	Indium–tin oxide (ITO) electrode	Genosensor	Electrochemical	DPV	Detection of *Salmonella* down to 10 genomes	-/60 min	[266]
Nanowires	Nickel nanowire bridge	Immunosensor	Electrochemical	EIS	Poultry meats/80 CFU mL^−1^	-/120 min	[184]
	Highly suspended carbon nanowires	Apasensor	Electrochemical	Conductivity	10 CFU mL^−1^	-/5 min	[267]
	Silicon nanowire	Genosensor	Electrochemical	Amperometric	Hybridisation of the DNA probe gives a current of 1.05 × 10^−10^A at 1 V compared to 5.86 × 10^−11^A at 1 V before hybridisation	-	[268]

* Analysis time (time to detection including sample preparation/detection).

## Data Availability

Not applicable.

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
