# Peer review of "Advancement in Salmonella Detection Methods: From Conventional to Electrochemical-Based Sensing Detection"

_biosensors, 2021, doi:10.3390/bios11090346_

Round 1
Reviewer 1 Report
This review summarizes the detection techniques for Salmonella, ranging from conventional to electrochemical techniques. I found this paper to be very valuable. I recommend this paper to be accepted for publication after minor revision.
- The Abstract section should be a little more detailed. The authors could have included more summary content.
- The content of Figure 2 makes perfect sense. But such a combination makes it impossible for the reader to see what is in it. I suggest the author to separate the diagrams in it. Or the author needs to increase the size of the font in the figure.
- Section 5 Miniaturisation of electrochemical biosensors. The author should have included pictures to illustrate the difference between miniaturized and conventional sensors.
Author Response
Dear Reviewer 1,
Thanks for fruitful comment on our article review. We are really appreciate the comment, then the correction was done based on those comments.
Kindly refer to attachment for our detail revised version.
Thanks

Reviewer 2 Report
The manuscript "Advancement in Salmonella Detection Method: From a Conventional to an Electrochemical-based Sensing Detection" by Awang et al. reviews the literature on biosensing in regard to Salmonella detection with short introduction/history to the salmonella caused diseases itself as motivation to then focus on biosensing of the pathogen and discussing in particular the developments in and around electrochemical sensing. The topic is for sure of interest and the literature overview given is indeed very helpful for readers starting into the field, thus I recommend publication in general but the manuscript suffers from a large deficit in language proficiency. It needs extensive (language) editing before publication. Not being native speaker myself, I might have overlooked many additional instances, but have marked what caught my attention in the uploaded PDF. After careful revision of these language issues (ideally proof read/edited for language by a native speaker) the review should be published in my opinion.

Author Response
Dear reviewer,
Thanks for fruitful comment for our review article. We are really appreciate the comment. The correction was done as commented on revise version.
Please go through revise version as attachment.
Thanks

Reviewer 3 Report
The article by M.S. Awang et al is an interesting review of the development of electrochemical detection techniques for Salmonella detection, however, a correction of this review paper is required before publication.
It was stated that: „An assay for Salmonella detection has been improved from a laborious bacterial culture method to a simple analytical biosensor, which can detect the presence of Salmonella or their cellular components in a short period of time” – Authors should directly specify the time needed for detection including the time of sample preparation from its collection for each discussed method.
In the introduction Section, it is not explained if this review is devoted to the Salmonella detection on species, strains, or serovars level. It should be this should be directly stated in this section.
In sections 1 and 3, the authors are describing conventional bacteria detection methods: biochemical assay, enzyme-linked immunosorbent assay (ELISA), polymerase chain reaction (PCR), however, it is known that owing to the genetic similarity among serovars, antibodies or nucleic acid probes show cross-reactions limiting the identification of Salmonella serovars. Therefore, the presented description of the state of art in the field of Salmonella detection is very poor and limited in my opinion as for the review paper. The authors describe well-known and well-established techniques in microbiological diagnostics but did not focus on describing the latest developments in bacterial detection. The authors should present a more recent description of the current advancements in the field of microbiological diagnostics. I am suggesting Authors read the articles related to other recent alternative methods of bacteria detection as:
Mass spectroscopy MALDI-TOF (see examples bellow):
https://doi.org/10.3390/foods10050933
https://doi.org/10.1016/j.foodcont.2020.107188
https://doi.org/10.1371/journal.pone.0040004
Optical phenotyping of bacteria colonies (see examples):
https://doi.org/10.1016/j.bios.2015.01.047
https://doi.org/10.1016/j.measurement.2021.109408
https://doi.org/10.1364/BOE.10.001165
https://doi.org/10.1117/1.JBO.21.10.107004
https://doi.org/10.1371/journal.pone.0135035
https://doi.org/10.1007/s00253-013-5495-4
https://doi.org/10.1128/mBio.01019-13
Optical fibers sensors (see examples):
https://doi.org/10.1016/j.foodcont.2015.09.031
https://doi.org/10.3390/s90705810
https://doi.org/10.1016/j.yofte.2018.09.012
Authors should describe a more recent state of the art in the field of bacteria sensing /detection as well as indicate the advantages of the electrochemical detection techniques in comparison with other recently developed alternatives (also these indicated above). These methods should be also indicated and described in sections (1,3) devoted to the recent techniques of bacteria detection.
In section 3 ( e.g. see Fig.1), the Authors distinguished the rapid methods of Salmonella detection, but they did not explain this statement. They should provide the appropriate time's limits for bacteria detection by described methods or its comparison in the table.
The authors in their review paper should describe in detail which input samples are used in the biosensors discussed. The environmental samples contain dozens of bacteria of different species, so they are heterogeneous samples. However, such samples cannot always be used by the discussed techniques and require additional sample preparation procedures - selection of representative colonies from agar plates and their multiplication. The authors should describe in detail which types of samples can be tested by which techniques/biosensors because, despite the speed of detection, the sample preparation process itself can increase the detection time so that it is comparable with standard microbiological techniques. Very often for purification of the initial sample, the bacteria colonies cultivation on agar plates is necessary Therefore the purification and bacteria cells multiplication processes can significantly increase the total time of the examination. Therefore, not only the time of detection but also the time of sample preparation is crucial for determining if the method is rapid or not. Please include such information in the review.
In section 4, it was stated that: ”There are numerous reports on Salmonella detection utilizing biosensor application in food safety research as well as in clinical diagnostics with a wide variety of sensing systems [6,8,111,112]. The next section will discuss in detail biosensor properties with an emphasis on the electrochemical biosensor.” – however, as was already indicated above authors use the term biosensor in a very selective context without taking into account the latest techniques, which in my opinion should not be the case for a review paper. The novel trends of bacteria detection should be also included and discussed by authors in this review.
In addition, the sensitivity of biosensors will depend on the minimal concentration of bacterial cells in the sample under test. However, the authors did briefly discuss this aspect in the paper. In my opinion, such information should also be included and compared with other alternative techniques.
As it was already indicated above, it is known that owing to the genetic similarity among serovars, antibodies or nucleic acid probes show cross-reactions limiting the identification of Salmonella serovars. However, this issue is briefly discussed here, which is a significant lack of this review in my opinion. Authors should indicate the techniques which or able to distinguish the Salmonella cells on the species, strains, and/or serovar level.
Can the Authors provide in Table 1 also the detection specificity of each method, because the information about sensitivity only is not sufficient? Please provide also information about kind of the input sample and the needed time for sample preparation from the initial sample.
The font should be larger in fig.2, Fig.6b.
Please, improve the resolution of figure 6b in the paper, because its quality is very low.
Author Response
Dear reviewer,
Thanks for fruiftul comment on our review article. We are really appreciate the comment. The article was revised based on the comment as attachment.
Please go through the attachment for detail correction.
Thanks

Round 2
Reviewer 3 Report
I congratulate the authors on the changes made, which have significantly improved the quality of their review work. In my opinion, the manuscript is suitable for publication in its present form.